# Enhanced Image Retrieval Using Multiscale Deep Feature Fusion in Supervised Hashing

**DOI:** 10.3390/jimaging11010020

**Published:** 2025-01-12

**Authors:** Amina Belalia, Kamel Belloulata, Adil Redaoui

**Affiliations:** 1High School of Computer Sciences, Sidi Bel Abbes 22000, Algeria; a.belalia@esi-sba.dz; 2RCAM Laboratory, Telecommunications Department, Sidi Bel Abbes University, Sidi Bel Abbes 22000, Algeria; kamel.belloulata@univ-sba.dz

**Keywords:** content-based image retrieval, hashing code, deep learning, multiscale feature extract, deep supervised hashing

## Abstract

In recent years, deep-network-based hashing has gained prominence in image retrieval for its ability to generate compact and efficient binary representations. However, most existing methods predominantly focus on high-level semantic features extracted from the final layers of networks, often neglecting structural details that are crucial for capturing spatial relationships within images. Achieving a balance between preserving structural information and maximizing retrieval accuracy is the key to effective image hashing and retrieval. To address this challenge, we introduce Multiscale Deep Feature Fusion for Supervised Hashing (MDFF-SH), a novel approach that integrates multiscale feature fusion into the hashing process. The hallmark of MDFF-SH lies in its ability to combine low-level structural features with high-level semantic context, synthesizing robust and compact hash codes. By leveraging multiscale features from multiple convolutional layers, MDFF-SH ensures the preservation of fine-grained image details while maintaining global semantic integrity, achieving a harmonious balance that enhances retrieval precision and recall. Our approach demonstrated a superior performance on benchmark datasets, achieving significant gains in the Mean Average Precision (MAP) compared with the state-of-the-art methods: 9.5% on CIFAR-10, 5% on NUS-WIDE, and 11.5% on MS-COCO. These results highlight the effectiveness of MDFF-SH in bridging structural and semantic information, setting a new standard for high-precision image retrieval through multiscale feature fusion.

## 1. Introduction

The surge in high-dimensional multimedia data, driven by advancements in computer networks and social media platforms, underscores the need for efficient storage and retrieval solutions [1,2,3,4,5]. Approximate Nearest Neighbor (ANN) searching [6] has emerged as a pivotal area of study in computer vision and information retrieval since it is a technique essential for reducing storage requirements and improving search efficiency in high-dimensional spaces. Hashing [7] has garnered considerable attention as a potent strategy within ANN searching, which transforms high-dimensional data into compact binary codes while preserving spatial relationships between data points. Deep hashing techniques [8,9,10,11] have been devised to concurrently learn visual features and binary hash codes, enriching encoded information with a semantic context. Deep hashing approaches have emerged as powerful tools for simultaneous feature learning and binary code generation. Unlike conventional hashing methods that rely on independently trained hash functions and quantization algorithms, deep hashing techniques adopt an end-to-end framework to extract semantic representations and construct binary hash codes. These end-to-end frameworks enable the cohesive construction of semantic representations and binary hash codes, surpassing traditional hashing techniques that rely on separate hash functions and quantization steps. While recent advancements in deep hashing have shown promise in information retrieval [12,13,14,15,16], there remains a need for further improvements, particularly in retaining local structural details within images. Most deep hashing approaches emphasize high-level features from fully connected (FC) layers [17,18,19], leading to a dearth of local feature information due to the global nature of these representations. Integrating multi-level features that capture both local and global details has shown promise for enhancing retrieval accuracy [20,21,22]. Despite attempting to fuse multi-feature representations, existing methods often fall short of achieving true end-to-end compatibility between feature representation and binary hash coding.

To address these limitations and to effectively harness the complementary nature of deep multi-scale features, this paper introduces Multiscale Deep Feature Fusion for High-Precision Image Retrieval through Supervised Hashing (MDFF-SH). By leveraging ResNet50 [23], convolutional multi-scale features are aggregated from images of varying sizes and fused within corresponding convolutional layers to yield robust representations. Inspired by the feature pyramid network [24], this fusion process incorporates top-down pathways and lateral connections, enabling the exploration of both top-layer semantic and bottom-layer spatial features. In summary, the key contributions of this paper are as follows:Dual-scale approach: We propose a dual-scale approach that considers both feature and image sizes to preserve the semantic and spatial details. Moreover, this compensates for the loss of high-level features and ensures the generated hash codes are more discriminative and informative.Multi-scale feature fusion: MDFF-SH learns hash codes across multiple feature scales and fuses them to generate final binary codes, enhancing retrieval performance.End-to-end learning: our MDFF-SH model integrates joint optimization for feature representation and binary code learning within a unified deep framework.Superior performance: extensive experiments on three well-known datasets demonstrated that MDFF-SH surpassed state-of-the-art approaches in retrieval performance.

## 2. Related Works

Hashing techniques have gained significant popularity in image retrieval due to their minimal storage requirements and fast processing capabilities [25,26]. The primary purpose of hashing is to map high-dimensional data into low-dimensional hash codes, ensuring that similar data points have minimal Hamming distances while dissimilar points have maximized distances.

Hashing methods are categorized into supervised [27,28] and unsupervised [29,30,31,32,33] approaches based on the use of labeled data. Researchers developed unsupervised hashing methods [34,35,36,37] to learn hash functions using unlabeled training samples, transforming input images into binary codes. Locality-Sensitive Hashing (LSH) [38] is one of the most well-known unsupervised methods, followed by significant approaches, such as Spectral Hashing (SH) [35] and Iterative Quantization (ITQ) [36].

In contrast, supervised hashing techniques leveraged labeled data to learn hash functions, often yielding higher accuracy compared with unsupervised methods. Supervised Hashing with Kernels (KSH) [39] employs kernel methods to create nonlinear hash functions. Minimal Loss Hashing (MLH) [40] uses structured SVMs to define an objective for learning hash functions. Supervised Discrete Hashing (SDH) [30] refines the objective function to produce high-quality hash codes without relaxation.

The emergence of deep neural networks propelled the development of deep hashing algorithms [14,18,41,42,43,44,45,46], which outperformed traditional methods by using rich feature representations. Researchers proposed pairwise and triplet-based similarity preservation strategies to utilize label information effectively. CNN-based Hashing (CNNH) [18] extracted features using CNNs but separated feature learning from hash function training, which limited feedback integration. Deep Pairwise-Supervised hashing (DPSH) [14] employed a Bayesian approach to model relationships between pairwise labels and hash codes, optimizing this relationship for better outcomes. HashGAN [41] uses a Wasserstein GAN to generate hash codes while leveraging pairwise similarities within a Bayesian framework. Zhuang et al. [42] developed a binary CNN classifier that leveraged triplet loss to maintain semantic relationships. Deep Triplet Quantization (DTQ) [43] integrated triplet-based quantization into a supervised learning framework, enabling the joint optimization of quantization and feature learning. In Supervised Semantics-Preserving Hashing (SSDH) [44], researchers embedded hash functions as a fully connected layer, focusing on minimizing the classification error during training. Wang et al. [45] offered a comprehensive framework for distance-preserving linear hashing extended to deep learning, where the fully connected layer’s features supported hashing. Shen et al.’s Similarity-Adaptive Deep Hashing (SADH) [46] utilized outputs from fully connected layers to refine a similarity graph matrix for enhanced hash code learning.

Product Quantization (PQ) techniques [34] have played a pivotal role in large-scale image retrieval due to their ability to compress high-dimensional data into compact representations while maintaining similarity-preserving properties. For instance, the study by Ma et al. [47] introduced a novel framework that leverages progressive quantization strategies to enhance fine-grained retrieval tasks. By integrating causal intervention into the quantization process, this approach achieves robust encoding and improved semantic preservation, making it highly effective for large-scale datasets. While vector quantization methods primarily optimize data compression and efficiency, our proposed MDFF-SH approach focuses on multiscale feature fusion to enhance the semantic representation and retrieval accuracy. These two paradigms are complementary, as future extensions of MDFF-SH could benefit from incorporating advanced quantization strategies to further improve the scalability in massive image databases.

Traditional approaches often relied on high-level features, typically from the final fully connected layers. However, capturing diverse features for a more comprehensive representation became a focus of multi-level image-retrieval methods. Lin et al. introduced Discriminative Deep Hashing (DDH) [16], which integrates end-to-end learning and multi-scale feature extraction from convolution-pooling layers. Yang et al. [48] developed Feature Pyramid Hashing (FPH), a dual-pyramid framework for learning detailed and semantic features for fine-grained retrieval. Redaoui et al. proposed Deep Feature Pyramid Hashing (DFPH) [49] to leverage multi-level visual and semantic data, and Deep Supervised Hashing with Multiscale Feature Fusion (DSHFMDF) [50], which extracts and combines multiscale features from various convolutional layers for robust image retrieval. Ng et al. [51] introduced Multi-Level Supervised Hashing (MLSH), which separately trains tables at different feature levels to enhance both the structural and semantic representations.

## 3. Proposed Methodology

This section outlines the proposed method in detail. First, we define the primary objective of our network: converting image information into hash codes for efficient retrieval. Next, we describe the structure of the proposed model, illustrated in Figure 1. Finally, we present the objective function that guides the optimization of the method.

### 3.1. Problem Definition

Let $X={\left\{x_{i}\right\}}_{i=1}^{N}\in R^{d\times N}$ represent a training dataset with *N* images, where $Y={\left\{y_{i}\right\}}_{i=1}^{N}\in R^{K\times N}$ are the associated ground truth labels for the $x_{i}$ samples, and *K* denotes the number of classes. To express the semantic similarities between images, we used a pairwise label matrix $S=\left\{s_{ij}\right\}$, where $s_{ij}\in \left\{0,1\right\}$ indicates whether images $x_{i}$ and $x_{j}$ are semantically related $s_{ij}=1$ or not $s_{ij}=0$. The objective of deep hashing is to learn a function $f:x\mapsto B\in {\left\{-1,1\right\}}^{L}$ that maps each input image $x_{i}$ to a binary code $b_{i}\in {\left\{-1,1\right\}}^{L}$, where *L* represents the length of the binary code.

### 3.2. Model Architecture

The architecture of the proposed MDFF-SH model, depicted in Figure 1, is structured to achieve high-efficiency and high-precision image retrieval through five main components: (1) feature extraction, (2) feature reduction, (3) feature fusion, (4) hash coding, and (5) classification. This modular approach ensures a cohesive understanding of each component and how they contribute to the model’s overall functionality.

**Feature extraction:** The initial feature extraction stage is crucial for gathering informative details from the input image. In MDFF-SH, the ResNet50 network serves as the backbone due to its capability to capture complex and distinguishing image features. Each layer in ResNet50 is designed to capture image details at increasing levels of abstraction, making it an ideal foundation for extracting both structural and semantic features. MDFF-SH systematically collects features from distinct levels of ResNet50. This includes low-level features, capturing fine details, such as edges and textures, and high-level features, encapsulating semantic attributes. This multi-level approach ensures that the image representation integrates both granular details and overall semantic meaning.**Multiscale feature focus:** The model’s multiscale feature extraction focuses on layers from several convolutional blocks—specifically, the final layers of the ‘conv3’, ‘conv4’, and ‘conv5’ blocks, along with the fully connected layer *fc1*. Lower-level layers, like ‘conv1’ and ‘conv2’, are excluded to optimize memory usage, as their semantic contribution is limited. The selected layers effectively capture a balanced mix of structural and semantic information, providing a comprehensive representation of the image that includes both low- and high-level characteristics.**Feature reduction:** After the extraction, the dimensionality of the multiscale features is reduced to retain discriminative power without excessive computational overhead. Using a 1 × 1 convolutional kernel, the model combines features across levels in a linear manner, creating a streamlined yet rich representation. This step enhances the depth and robustness of the features while minimizing the redundancy.**Feature fusion:** In the fusion stage, the reduced features from different levels are combined to produce a unified representation. By merging both low- and high-level information, the fusion layer enables the model to construct an image representation that captures local structures alongside the global context. This fusion provides a robust basis for generating binary codes that reflect a detailed and semantically rich image profile.**Hash coding:** To generate the final hash codes, the fused feature representation undergoes nonlinear transformations through hash layers, each of which outputs binary codes of the desired length *L*. This transformation ensures that the binary codes retain the core characteristics of the images in a compact and retrieval-optimized format.**Classification:** The classification layer, which corresponds to the number of classes in the dataset, assigns the generated hash codes to specific image categories. This final component allows MDFF-SH to distinguish between classes based on learned binary representations, reinforcing the network’s retrieval effectiveness.

Through this structured architecture presented in Table 1, MDFF-SH captures both the local and global image information, resulting in a powerful and compact feature representation that is tailored to high-precision image retrieval.

After extracting features from multiple scales, we employed a 1 × 1 convolutional kernel to reduce the dimensionality while preserving the discriminative information. This process enhances the feature depth and robustness and eliminates redundancy.

Subsequently, a fusion layer composed of 1024 nodes integrates these multi-scale features, combining low-level structural details with high-level semantic information. This fusion step creates a comprehensive image representation that balances fine-grained local structures with broader contextual understanding.

To generate compact binary hash codes, we applied a nonlinear mapping through multiple hash layers, each with L nodes. This nonlinear transformation effectively encapsulates key image characteristics into binary codes. The concatenated hash code representation is further refined in the final hashing layer to ensure semantic integrity and discriminative power.

Finally, a classification layer with neurons corresponding to the number of classes is employed to categorize images based on their learned representations. The discriminative nature of the hash codes enables accurate image classification.

By integrating multi-scale features and a well-structured architecture, our model generates diverse and informative hash codes. These hash codes effectively capture both the local details and global context, leading to improved retrieval performance and accurate image classification.

### 3.3. Loss Functions and Learning Rule

To ensure that the generated hash codes effectively preserve semantic similarity, our MDFF-SH method combines three distinct loss functions: pairwise similarity loss, quantization loss, and classification loss. These losses are harmonized to support efficient and effective training.

#### 3.3.1. Pairwise Similarity Loss

The MDFF-SH method is designed to maintain similarity between pairs of input samples within Hamming space. Pairwise similarity is evaluated by calculating the inner product between the hash codes $b_{i}$ and $b_{j}$, defined as dist $dist_{H}\left(b_{i},b_{j}\right)=\frac{1}{2}b_{i}^{T}b_{j}$. Given a set of binary codes $B={\left\{b_{i}\right\}}_{i=1}^{N}$ and pairwise labels $S=\left\{s_{ij}\right\}$, the probability of the pairwise labels is represented as
(1)$$p\left(s_{ij}|B\right)=\left\{\begin{array}{cc} \sigma (w_{ij}) & \mathrm{if}\;s_{ij}=1\\ 1-\sigma (w_{ij}) & \mathrm{if}\;s_{ij}=0 \end{array}\right.$$
where $\sigma \left(w_{ij}\right)=\frac{1}{1+e^{-w_{ij}}}$ and $w_{ij}=\frac{1}{2}b_{i}^{T}b_{j}$

This formulation implies that a larger inner product $\langle b_{i},b_{j}\rangle$ corresponds to a smaller $dist_{H}\left(b_{i},b_{j}\right)$ and a higher value of $p\left(1|b_{i},b_{j}\right)$. When $s_{ij}=1$, the binary codes $b_{i}$ and $b_{j}$ are considered similar.

The optimization problem then becomes minimizing the negative log-likelihood over labels in *S*, resulting in
(2)$$J_{1}=-\log p\left(S|B\right)=-\sum_{s_{ij}\in S}\left(s_{ij}w_{ij}-\log\left(1+e^{w_{ij}}\right)\right)$$

This objective function aims to minimize the Hamming distance between similar samples while maximizing the distance between dissimilar samples, aligning with the principles of similarity-based hashing.

#### 3.3.2. Quantization Loss

In practical applications, binary hash codes are commonly used for measuring similarity. However, optimizing discrete hash codes directly within a neural network can be challenging. To address this, we employed a continuous approximation for hash coding. Let $u_{i}$ denote the output of the hash layer, with $b_{i}$ defined as $b_{i}=\mathrm{sgn}\left(u_{i}\right)$. To minimize the gap between continuous and discrete representations, we introduced quantization loss as a secondary objective:(3)$$J_{2}=\sum_{i=1}^{Q}{\left|b_{i}-u_{i}\right|}_{2}^{2}$$
where *Q* represents the mini-batch size.

#### 3.3.3. Classification Loss

To support the robust learning of multiscale features across the network, we employ cross-entropy loss for classification, which helps the model correctly categorize input samples. The classification loss is given by
(4)$$J_{3}=-\sum_{i=1}^{Q}\sum_{k=1}^{K}y_{i,k}\log\left(p_{i,k}\right)$$
where $y_{i,k}$ denotes the true label and $p_{i,k}$ represents the softmax output of the *i*-th training sample for the *k*-th class.

In conclusion, the total loss function combines the pairwise similarity, quantization, and classification losses, as follows:(5)$$J=J_{1}+\beta J_{2}+\gamma J_{3}$$
where β and γ are balancing parameters that control the contributions of the quantization and classification losses, respectively.

## 4. Experiments

This section evaluates the performance of MDFF-SH and its variations on three extensive public datasets: CIFAR-10, NUS-WIDE, and MS-COCO. Our objective was to demonstrate the effectiveness of the proposed method compared with several leading hashing approaches. We begin with an overview of these datasets and follow with the experimental setup. Section 4.3 details the evaluation metrics and baseline methods. Finally, we present the results, including a comparative analysis with state-of-the-art hashing techniques.

### 4.1. Datasets

**CIFAR-10** [52]: This dataset consists of 60,000 color images across ten object classes, with each class containing 6000 images sized 32 × 32 pixels. Following the protocol in [53], we randomly selected 1000 images (100 per class) as the query set, with the remaining images serving as the database. From the database, we sampled 5000 images (500 per class) as the training set.

**NUS-WIDE** [54]: Comprising 123,287 color images (40,504 validation images and 82,783 training images), this dataset includes images labeled with one or more of 80 categories. For our experiments, we randomly selected 5000 images as the query set, 10,000 as the training set, and used the remaining images as the database.

**MS-COCO** [55]: This is a dataset consisting of 123,287 color images (40,504 validation and 82,783 training). Each image is labeled by one or more of 80 categories. We randomly selected 5000 images as query points and 10,000 images as the training dataset. The rest of the images were used as the database.

### 4.2. Experimental Settings

The MDFF-SH method was implemented using the PyTorch 2.0 deep learning framework, and we initialized the network parameters with the ResNet50 convolutional model pretrained on the ImageNet dataset [56]. All experiments were conducted using the RMSProp optimizer [57] with a learning rate of 1 × 10^−5^ and a batch size of 32. The hyperparameters were set as follows: γ=0.01 and β=0.1.

### 4.3. Evaluation Metrics and Baselines

To assess the performance of our image retrieval method and facilitate comparisons with alternative approaches, we used the following metrics:Mean Average Precision (MAP) results;Precision–recall (PR) curves;Precision at top retrieval levels (P@N);Precision within a Hamming radius of 2 (P@H ≤ 2).

The MDFF-SH method was compared with a selection of traditional and state-of-the-art methods, including five unsupervised methods: LSH [6], SH [35], SGH [58], ITQ [36], and PCAH [59], as well as two supervised hashing methods: SDH [30] and KSH [39]. Additionally, we included nine deep supervised hashing methods: CNNH [18], DNNH [10], DCH [60], DHN [9], HashNet [61], DHDW [62], DPH [63], LRH [53], and MFLH [64]. For multi-label datasets, such as MS-COCO and NUS-WIDE, two samples were considered similar if they shared one or more semantic label.

### 4.4. Results

Table 2 presents a comparison of the MAP results for our method and competing hashing methods on CIFAR-10 and NUS-WIDE with hash code lengths of 12, 24, 32, and 48 bits. Our MDFF-SH method consistently outperformed all the other methods. Specifically, compared with the best traditional hashing method SDH [30], MDFF-SH achieved average MAP improvements of 52.7% and 23.55% on CIFAR-10 and NUS-WIDE, respectively. Deep hashing methods generally perform better than classical methods due to their ability to generate more robust feature representations. For CIFAR-10 and NUS-WIDE, MDFF-SH delivered average MAP increases of 9.58% and 2.95%, respectively, over the second-best method MFLH [64] across all hash code lengths. For example, the MAP values of MDFF-SH at different lengths were enhanced by 52.6%, 52.5%, 53.3%, and 52.4%, respectively, compared with the SDH method. The MAP of the MDFF-SH method was also significantly improved compared with the deep hash method. These results indicate the capability of MDFF-SH to produce high-quality hash codes for efficient image retrieval.

Table 3 shows the performance of MDFF-SH on the MS-COCO dataset. MDFF-SH achieved a superior retrieval performance at all code lengths compared with all the baseline methods. As a multi-label dataset, MS-COCO presented a more complex semantic structure than the single-label datasets, which posed a greater challenge for maintaining semantic integrity in hash codes. For example, the MDFF-SH method improved the MAP values over different lengths of hash codes by 7.4%, 11.7%, 14.1%, and 15.2%, respectively, compared with the DCH method. Nonetheless, MDFF-SH achieved the best results, underscoring the effectiveness and robustness of the proposed approach for high-precision image retrieval in complex datasets.

Figure 2a and Figure 3a present the precision curves for P@H = 2, demonstrating that our MDFF-SH method consistently outperformed other techniques by achieving the highest precision within this Hamming radius. Although a slight decline in the P@H = 2 performance was observed as the code length increased, MDFF-SH maintained a strong retrieval accuracy, indicating its ability to focus on relevant points within a Hamming radius of 2, even with longer hash codes.

Additionally, Figure 2b,c and Figure 3b,c compare the precision–recall and precision at top results performance of MDFF-SH with the other methods. In particular, Figure 2c and Figure 3c show that MDFF-SH achieved the highest precision with 48-bit codes across varying numbers of returned samples, especially in the range of 100 to 1000. Furthermore, Figure 2b and Figure 3b illustrate that MDFF-SH achieved notably high precision at low recall levels—a crucial feature for precision-first retrieval systems widely used in practical applications. Overall, these results underscore the superior performance of MDFF-SH compared with the other methods evaluated.

Figure 2a and Figure 3a display the precision curves for P@H = 2, clearly showing that our MDFF-SH method outperformed other approaches by achieving the highest precision within this Hamming radius. While a slight decrease in the P@H = 2 performance occurred as the code length increased, this result highlights MDFF-SH’s effectiveness in concentrating on relevant points within a Hamming radius of 2, even with longer hash codes.

In summary, our MDFF-SH method consistently outperformed the compared methods across various evaluation metrics, underscoring its superiority in image retrieval tasks. To visually illustrate its effectiveness in eliminating irrelevant images, we present Figure 4, showcasing the retrieval accuracy of different image categories in the CIFAR-10 dataset using MDFF-SH with 48-bit binary codes. This figure features query images in the first column, while the subsequent columns display images retrieved using MDFF-SH. This example reinforced our approach’s capability to precisely retrieve pertinent images, further substantiating its practical utility.

In summary, our MDFF-SH method consistently surpassed the compared techniques across multiple evaluation metrics, affirming its superior performance in image retrieval tasks. To visually demonstrate its effectiveness in filtering out irrelevant images, Figure 4 presents the retrieval accuracy for various image categories within the CIFAR-10 dataset using MDFF-SH with 48-bit binary codes. In this figure, query images are shown in the first column, followed by images retrieved through MDFF-SH in the subsequent columns. This example highlights our method’s ability to accurately retrieve relevant images, underscoring its practical value in real-world applications.

### 4.5. Ablation Studies

(1)Ablation studies on multi-level image representations for enhanced hash learning: To investigate the impact of multi-level image representations on hash learning, we conducted ablation studies. Unlike many existing methods that primarily focus on semantic information extracted from the final fully connected layers, we explored the contribution of structural information from various network layers. Table 4 presents the retrieval performance on the CIFAR-10 dataset using different feature maps. We observed that features from the *fc*1 layer yielded the highest MAP of 75.8%, emphasizing the importance of high-level semantic information. However, using features from convs 3–5 resulted in an average MAP of 62.5%, highlighting the significance of low-level structural details. Our proposed MDFF-SH approach outperformed all other configurations, where it achieved an average MAP of 85.5%, and thus, demonstrated the effectiveness of combining multi-scale features for enhanced retrieval performance.(2)Ablation studies on the objective function: To assess the impact of different loss components in our objective function, we conducted ablation studies on the CIFAR-10 dataset using the MDFF-SH model. We evaluated the performance of the model when either the pairwise quantization loss (β=0, MDFF-SH-J3) or the classification loss (γ=0, MDFF-SH-J2) was excluded. As shown in Table 5, the inclusion of both J2 and J3 resulted in an 8.55% performance improvement. This finding highlights the importance of both the quantization loss, which minimizes the quantization error, and the classification loss, which preserves semantic information, for generating high-quality hash codes.

## 5. Conclusions and Future Work

This paper introduces a novel end-to-end framework, Multiscale Deep Feature Fusion for High-Precision Image Retrieval through Supervised Hashing (MDFF-SH), designed to generate robust binary codes. Our approach optimizes three key components: similarity loss, quantization loss, and semantic loss, to effectively integrate structural information into hash representations. By leveraging multiscale features, MDFF-SH achieves a balance between the structural detail and retrieval accuracy, leading to improved recall and precision.

Extensive experiments on standard image retrieval datasets demonstrate the superior performance of MDFF-SH compared with state-of-the-art methods. In future work, we aim to extend this approach to medical imaging, where the presence of multi-scale objects could benefit significantly from our method’s ability to capture both fine-grained and coarse-grained details.

The scalability of our model makes it adaptable to various computer vision tasks, providing robust feature representations that have the potential to advance a wide range of applications.

## Figures and Tables

**Figure 1 jimaging-11-00020-f001:**
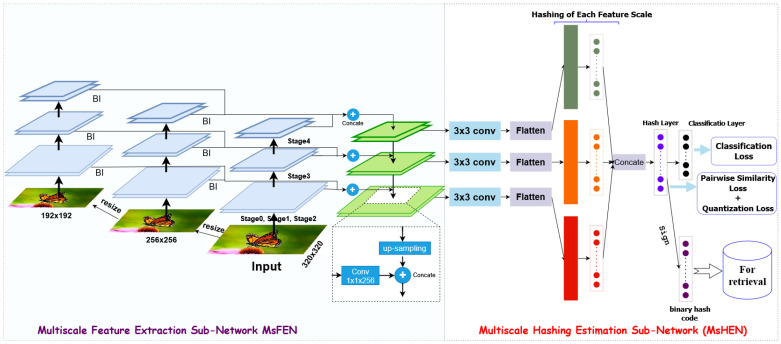
Enhanced image retrieval using Multiscale Deep Feature Fusion in Supervised Hashing (MDFF-SH).

**Figure 2 jimaging-11-00020-f002:**
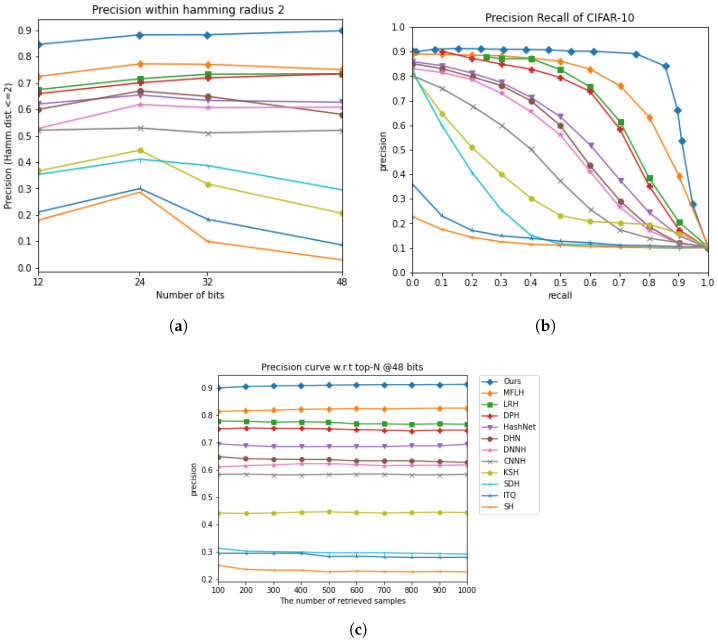
The comparison results on the CIFAR-10 dataset under three evaluation metrics. (**a**) Precision within Hamming radius 2. (**b**) Precision recall curve on 48 bits. (**c**) Precision curve with respect to top-N @48 bits.

**Figure 3 jimaging-11-00020-f003:**
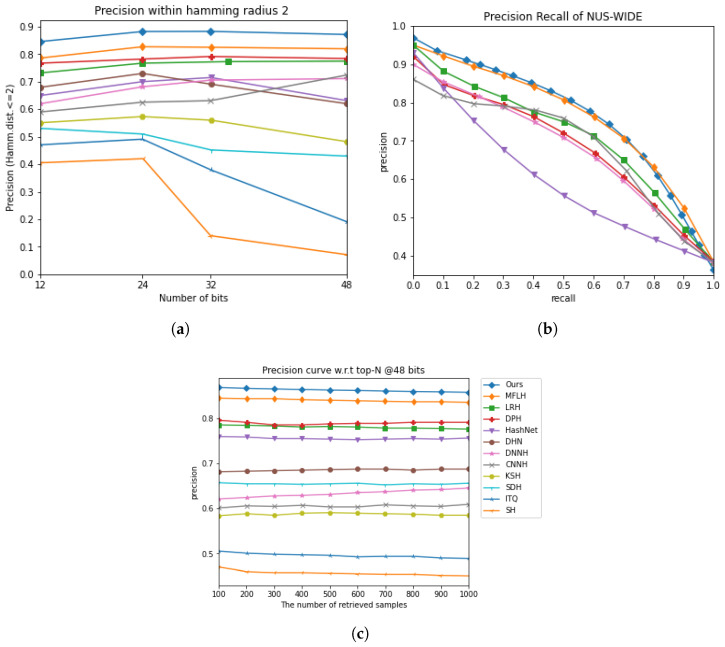
The comparison results on the NUS-WIDE dataset under three evaluation metrics. (**a**) Precision within Hamming radius 2. (**b**) Precision recall curve on 48 bits. (**c**) Precision curve with respect to top-N @48 bits.

**Figure 4 jimaging-11-00020-f004:**
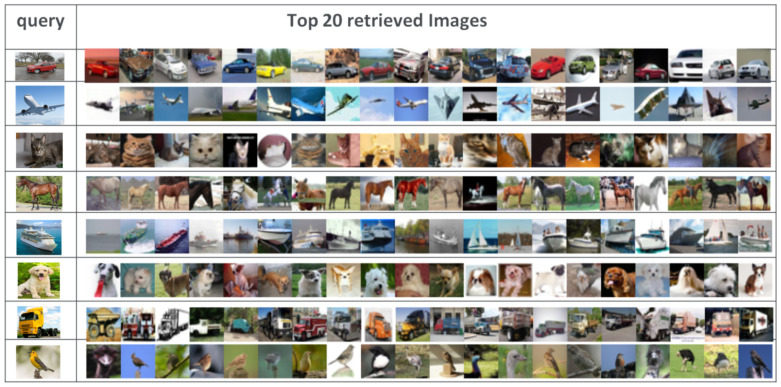
Presented are the top 20 retrieved results from the CIFAR-10 dataset, which utilized MDFF-SH with 48-bit hash codes. The first column showcases the query images, while the subsequent columns display the retrieval results generated by MDFF-SH.

**Table 1 jimaging-11-00020-t001:** Summary of the feature extraction network. Layers marked with ‘#’ are used for feature extraction. ReLU and batch normalization layers are omitted for simplicity.

Conv Block	Layers	Kernel Sizes	Feature Dimensions
1	Conv2D, Conv2D#, MaxPooling	64×3×3, 64×3×3	224×224
2	Conv2D, Conv2D#, MaxPooling	128×3×3, 128×3×3	112×112
3	Conv2D, Conv2D, Conv2D, Conv2D#, MaxPooling	256×3×3, 256×3×3, 256×3×3, 256×3×3	56×56
4	Conv2D, Conv2D, Conv2D, Conv2D#, MaxPooling	512×3×3, 512×3×3, 512×3×3, 512×3×3	28×28
5	Conv2D, Conv2D, Conv2D, Conv2D#, MaxPooling	512×3×3, 512×3×3, 512×3×3, 512×3×3	14×14

**Table 2 jimaging-11-00020-t002:** Mean Average Precision (MAP) of the Hamming ranking for different numbers of bits on CIFAR-10 and NUS-WIDE. The MAP values were calculated for the top 5000 retrieval images from the NUS-WIDE dataset.

Method	CIFAR-10 (MAP)				NUS-WIDE (MAP)			
	**12 Bits**	**24 Bits**	**32 Bits**	**48 Bits**	**12 Bits**	**24 Bits**	**32 Bits**	**48 Bits**
SH [35]	0.127	0.128	0.126	0.129	0.454	0.406	0.405	0.400
ITQ [36]	0.162	0.169	0.172	0.175	0.452	0.468	0.472	0.477
KSH [39]	0.303	0.337	0.346	0.356	0.556	0.572	0.581	0.588
SDH [30]	0.285	0.329	0.341	0.356	0.568	0.600	0.608	0.637
CNNH [18]	0.439	0.511	0.509	0.522	0.611	0.618	0.625	0.608
DNNH [10]	0.552	0.566	0.558	0.581	0.674	0.697	0.713	0.715
DHN [9]	0.555	0.594	0.603	0.621	0.708	0.735	0.748	0.758
HashNet [61]	0.609	0.644	0.632	0.646	0.643	0.694	0.737	0.750
DPH [63]	0.698	0.729	0.749	0.755	0.770	0.784	0.790	0.786
LRH [53]	0.684	0.700	0.727	0.730	0.726	0.775	0.774	0.780
MFLH [64]	0.726	0.758	0.771	0.781	0.782	0.814	0.817	0.824
**MDFF-SH**	**0.811**	**0.854**	**0.874**	**0.880**	**0.828**	**0.854**	**0.866**	**0.887**

**Table 3 jimaging-11-00020-t003:** Mean Average Precision (MAP) of the Hamming ranking for different numbers of bits on MS-COCO. The MAP values were calculated for the top 5000 retrieval images.

Method	MS-COCO (MAP)			
	**16 Bits**	**32 Bits**	**48 Bits**	**64 Bits**
SGH [58]	0.362	0.368	0.375	0.384
SH [35]	0.494	0.525	0.539	0.547
PCAH [59]	0.559	0.573	0.582	0.588
LSH [6]	0.406	0.440	0.486	0.517
ITQ [36]	0.613	0.649	0.671	0.680
DHN [9]	0.608	0.640	0.661	0.678
HashNet [61]	0.642	0.671	0.683	0.689
DCH [60]	0.652	0.680	0.689	0.690
DHDW [62]	0.655	0.681	0.695	0.702
**MDFF-SH**	**0.726**	**0.797**	**0.830**	**0.842**

**Table 4 jimaging-11-00020-t004:** Mean Average Precision (MAP) for different feature scales with various bit lengths on CIFAR-10.

Method	CIFAR-10 (MAP)			
	**12 Bits**	**24 Bits**	**32 Bits**	**48 Bits**
fc1	0.710	0.761	0.775	0.788
Convs3–5	0.580	0.595	0.639	0.688
MDFF-SH	0.811	0.854	0.874	0.880

**Table 5 jimaging-11-00020-t005:** Mean Average Precision (MAP) results for different variants of the objective function on CIFAR-10.

Method	CIFAR-10 (MAP)			
	**12 Bits**	**24 Bits**	**32 Bits**	**48 Bits**
MDFF-SH-J2	0.667	0.812	0.830	0.852
MDFF-SH-J3	0.656	0.742	0.785	0.796
MDFF-SH	0.811	0.854	0.874	0.880

## Data Availability

Publicly available datasets were analyzed in this study. These data can be found here: http://www.cs.toronto.edu/~kriz/cifar.html, https://paperswithcode.com/datasets (all accessed on 31 July 2024).

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
