# Peer review of "Enhanced Image Retrieval Using Multiscale Deep Feature Fusion in Supervised Hashing"

_2313-433X, 2025, doi:10.3390/jimaging11010020_

Round 1

Reviewer 1 Report

Comments and Suggestions for Authors

1. The proposed method can be compared with graph convolutional network (GCN) based methods such as GCH and GCN-H. Compared with these methods, the proposed method can demonstrate its advantages in multi-scale feature fusion.

2.In the abstract, it is mentioned that "the balance between structural information and image retrieval accuracy is the key to image hashing and retrieval", but the article does not explain how this balance is achieved.

3.In the last sentence of the first paragraph of "Introduction", the word "attempts" should be replaced with "attempting". In the first contribution proposed, it is better to replace "that" with a clear subject, and it would be more appropriate to add "that" after "ensures".

4.The content of the relevant work section should maintain consistency in the predicate verb tense of each sentence.

5.Figure 1 is an introduction to the proposed method, which should be more detailed, and the text in the figure is too small, with overlapping lines and text.

6.The literature review is unsatisfactory,. About the semantic hashing, some recent references propose different similarity preserving metrics should be mentioned in the manuscript. In addtion, related quantization methods for large-scale image retrieval such as [1] should discussed in the manuscript. [1] Deep Progressive Asymmetric Quantization Based on Causal Intervention for Fine-Grained Image Retrieval, tmm'2024

Comments on the Quality of English Language

1. In the last sentence of the first paragraph of "Introduction", the word "attempts" should be replaced with "attempting". In the first contribution proposed, it is better to replace "that" with a clear subject, and it would be more appropriate to add "that" after "ensures".

2.The content of the relevant work section should maintain consistency in the predicate verb tense of each sentence.

Reviewer 2 Report

Comments and Suggestions for Authors

1. So what is hashing? can't sure it's given adequately discussed in the introduction

2. you first said conventional and deep hashing but than divide into 2 as supervised and unsupervised. very confusing?

3. table 2 and table 3: why not using same methods? why different bits? also cpu time comparison should be added. some statistical analysis are also be meaningful

4. also still can't learn properly the importance of hashing after reading the papers. 

Reviewer 3 Report

Comments and Suggestions for Authors

Image retrieval method is presented, that incorporates representing images in several scales and hashing. The method contains some novelty and is reported to have better quality compared to rivals in experiments with known benchmark datasets. The paper gas some important drawbacks.

1. The abstract contains only words in general, but not a description of distinctive features of method. For instance, what is the difference, compared to [14], which also contains multiscale representation and hashing? The hallmark of the method should be presented in the abstract.

2. Section 2.1 is not actually problem definition. There should be no "training dataset" in problem. Training dataset appears only when and if the developer decides to use machine learning (ML) methods to solve the problem. However non-trainable methods can be applied without any "training dataset". So, Section 2.1 concerns in fact the method of solution. Problem statement should be given purely.

3. Further on Section 2.1. As a method description it is not completed. Presumably, as typical for ML tasks, it might be written as optimization problem in form of some functional to be minimized upon some arguments and some bounding conditions. There is no such formulation, it should be given.

4. Line 178. Some 'nonlinear transformation' is mentioned. What is it in detail? What are these 'multiple hash layers'. It should be described.

5. Line 197: dist_H(b_i,b_j) is introduced. Line 200 w_{ij} is introduced by same formula. What is the difference? What is the need to introduce two denotations of the same thing? Please explain.

6. Lines 235-248 and Table 2. It seems that each of the databases was split into training and test sets once and one training-classification experiment was carried out. This is not sufficient. Cross-validation (i.e. several experiments with varying partitioning) should be used. First, this gives more precise estimation of the target value (MAP), second, it allows to estimate dispersion.

7. Line 254. Letter \beta clashes with same letter denoting quantization loss weight in formula (5).

8. Matching with rivals shows superiority of the proposed method in precision. However, this picture is incomplete without matching the count of trainable parameters and the processing time (if possible). More trainable parameters give better precision, but may produce slower inference, so the method might not be the best choice in every application. Number of trainable parameters should be given.

Given drawbacks should be corrected, then the paper should be reviewed once more.

Round 2

Reviewer 1 Report

Comments and Suggestions for Authors

The vector quantization methods should be discussed in the related work. It plays an important role in large-scale image retrieval. Recent research trends of  vector quantization have been presented in DOI: 10.1109/TMM.2024.3407661.  The authors should give some disussions in the related work.

Author Response

Comment: "The vector quantization methods should be discussed in the related work. It plays an important role in large-scale image retrieval. Recent research trends of vector quantization have been presented in DOI: 10.1109/TMM.2024.3407661. The authors should give some discussions in the related work."

Response:
Thank you for your insightful suggestion. We agree that vector quantization methods are fundamental to large-scale image retrieval, and discussing them would provide valuable context for our work.

  1. Inclusion of Vector Quantization Methods:

    • In the revised manuscript, we have expanded the "Related Works" section (Line 101-112) to include a discussion on vector quantization techniques and their significance in image retrieval.
    • The added paragraph: Product Quantization (PQ) techniques [34 ] have played a pivotal role in large-scale image retrieval due to their ability to compress high-dimensional data into compact representations while maintaining similarity-preserving properties.  For instance, the study by Ma et al. [47] introduced a novel framework that leverages progressive quantization strategies to enhance fine-grained retrieval tasks. By integrating causal intervention into the quantization process, this approach achieves robust encoding and improved semantic preservation, making it highly effective for large-scale datasets. While vector quantization methods primarily optimize data compression and efficiency, our proposed MDFF-SH approach focuses on multiscale feature fusion to enhance semantic representation and retrieval accuracy. These two paradigms are complementary, as future extensions of MDFF-SH could benefit from incorporating advanced quantization strategies to further improve scalability in massive image databases.
  2. Incorporation of the Suggested Reference:

    • We have added the referenced paper (DOI: 10.1109/TMM.2024.3407661) [47] and incorporated the relevant insights into the discussion. This helped us highlight recent advancements in vector quantization and its relevance to our approach.
  3. Relevance to Our Work:

    • The revised section (Related Works Line 101-112) will also address how vector quantization methods compare to and complement the multiscale feature fusion strategy proposed in our work, thereby situating our method within the broader research landscape.

Reviewer 3 Report

Comments and Suggestions for Authors

The authors have adequately corrected and explained all issues from the previous comments. The paper can be published now.

Author Response

Thank you for your positive feedback and for acknowledging the revisions made in response to the previous comments. We greatly appreciate your thorough review and constructive suggestions, which have significantly contributed to improving the quality of our manuscript.

We are pleased that the revised version meets your expectations, and we are grateful for your recommendation for publication.